# Functional Role of RING Ubiquitin E3 Ligase VdBre1 and VdHrd1 in the Pathogenicity and Penetration Structure Formation of *Verticillium dahliae*

**DOI:** 10.3390/jof9101037

**Published:** 2023-10-21

**Authors:** Xing Yang, Zhijuan Hu, Jingjie Yuan, Run Zou, Yilan Wang, Xuan Peng, Shan Xu, Chengjian Xie

**Affiliations:** 1The Chongqing Key Laboratory of Molecular Biology of Plant Environmental Adaptations, Chongqing Normal University, Chongqing 401331, China; 18875051413@163.com (X.Y.); y18225267527@163.com (J.Y.); 20200017@cqnu.edu.cn (S.X.); 2Chongqing Engineering Research Center of Specialty Crop Resources, The College of Life Science, Chongqing Normal University, Chongqing 401331, China

**Keywords:** *Verticillium dahliae*, ubiquitin ligase E3, histone ubiquitination, VdBre1, VdHrd1

## Abstract

*Verticillium dahliae*, a virulent soil-borne fungus, elicits Verticillium wilt in numerous dicotyledonous plants through intricate pathogenic mechanisms. Ubiquitination, an evolutionarily conserved post-translational modification, marks and labels proteins for degradation, thereby maintaining cellular homeostasis. Within the ubiquitination cascade, ubiquitin ligase E3 demonstrates a unique capability for target protein recognition, a function often implicated in phytopathogenic virulence. Our research indicates that two ubiquitin ligase E3s, VdBre1 and VdHrd1, are intrinsically associated with virulence. Our findings demonstrate that the deletion of these two genes significantly impairs the ability of *V. dahliae* to colonize the vascular bundles of plants and to form typical penetration pegs. Furthermore, transcriptomic analysis suggests that VdBre1 governs the lipid metabolism pathway, while VdHrd1 participates in endoplasmic-reticulum-related processes. Western blot analyses reveal a significant decrease in histone ubiquitination and histone H3K4 trimethylation levels in the *ΔVdBre1* mutant. This research illuminates the function of ubiquitin ligase E3 in *V. dahliae* and offers fresh theoretical perspectives. Our research identifies two novel virulence-related genes and partially explicates their roles in virulence-associated structures and gene regulatory pathways. These findings augment our understanding of the molecular mechanisms inherent to *V. dahliae*.

## 1. Introduction

The ubiquitination mechanism involves the consecutive actions of ubiquitin activating enzyme (E1), ubiquitin conjugating enzyme (E2), and ubiquitin ligase (E3), which attach ubiquitin to a target protein [1]. The high specificity and selectivity of the UPS system stem from the diversity of ubiquitin-protein ligase E3s capable of recognizing target proteins specifically. Protein turnover is a fundamental mechanism in the regulation of cellular functions in eukaryotes, underpinned by two major proteolytic systems: the ubiquitin-proteasome system (UPS) and lysosomes [2]. The UPS predominantly regulates the non-lysosomal proteolysis of intracellular proteins, playing a central role in controlling cellular processes including apoptosis, cell cycle progression, differentiation, and cell trafficking. In fungi such as *Cryphonectria parasitica*, the cpubi4 polyubiquitin gene is associated with the regulation of growth, spore production, and pathogenicity, thereby controlling the ubiquitination of critical metabolic proteins [3].

Two primary categories of E3s have been identified, specifically the RING (really interesting new gene) domain-containing E3s [4] and HECT (homologous to the E 6-AP carboxyl terminus) domain E3s [5], which play pivotal roles in enabling the transfer of ubiquitin from E2 enzymes to their substrates. RING E3s orchestrate this transfer process by directly conveying ubiquitin from E2 enzymes to substrate proteins, thereby acting as a bridge between the E2 enzymes and substrate proteins [6]. On the other hand, HECT E3s take a slightly different approach by forming an intermediary thioester bond between the active site cysteine within their HECT domain and the ubiquitin obtained from E2, prior to its transfer to the intended substrate [7]. This detail underscores the contrasting methods employed by the two major E3 classifications to facilitate the essential process of ubiquitin transfer from E2 enzymes to substrates.

Existing research on E3s in pathogenic fungi demonstrates the critical role of many E3s in fungal virulence. Studies on *Saccharomyces cerevisiae* homologs of Grr1 and Cdc4, the two most widely researched SCF ligases, have revealed their involvement in nutrient responses and cell cycle regulation [8]. Similarly, in *Magnaporthe oryzae*, MoGrr1, an ortholog of Grr1, has been connected to conidial development, cell wall integrity, and virulence [9]. Expression studies have suggested the up-regulation of *GrrA* during the developmental stages of cleistothecial formation in *Aspergillus nidulans*. Additionally, it was found that GrrA plays a vital role in the processes of meiosis and sexual spore formation within *A. nidulans* [10]. Cullin-RING ligases (CRLs) represent the most extensive class of ubiquitin ligases. CRL complexes comprise four essential components: a cullin scaffold protein, an adaptor protein, a substrate recognizing receptor, and a RING protein serving as the catalytic subunit [11]. A number of F-box proteins, crucial constituents of the SCF protein E3 ubiquitin ligase complex, such as Frp1 in *Fusarium oxysporum* [12] and MoFwd1, MoCDc4, MoFbx15, and Hect type E3Upl3 in *M. oryzae*, have been ascertained to be vital for fungal virulence [13,14,15]. In *Neurospora crassa*, the F-box protein MUS-10 plays a crucial role in the control of mitochondrial maintenance and cellular senescence [16]. In the human pathogen *Cryptococcus neoformans*, the F-box protein Fbp1 is essential for fungal sporulation and virulence [17,18]. In the plant pathogen *F. oxysporum*, the F-box protein Fbp1 regulates virulence and invasive growth via the mitogen-activated protein kinase (MAPK) pathway [19]. A total of 51 and 55 F-box genes were identified in *A. fumigatus* and *A. flavus*, respectively [11]. The necessity of Fbx23 and Fbx47 has been demonstrated in relation to carbon catabolite repression [20], while Fbx15 is required for the positive regulation of both asexual and sexual development [21]. Fbx15 also plays a crucial role in regulating virulence and the response to oxidative stress [22]. Moreover, Bre1, a notably conserved E3 in *S. cerevisiae*, is essential in the monoubiquitination of histone H2B, and its counterpart, AaBre1 in *Alternaria alternata*, has been found to regulate fungal pathogenicity and vegetative growth through its influence on H2B monoubiquitination [23]. Hrd1p, a yeast ERAD ubiquitin ligase E3, and Der1p, a multispanning membrane protein, are reportedly involved in the ubiquitination and transport of proteins with defective folding structures. In *M. oryzae*, their analogous counterparts, MoHrd1 and MoDer1, are associated with variations in appressorium morphology, pathogenicity, and sensitivity to endoplasmic reticulum stress [24]. 

Verticillium wilt, caused by the soil-borne pathogen *V. dahliae*, is a destructive disease that affects over two hundred dicotyledonous plants [25]. Due to its capability to colonize the xylem and its persistent resting structure microsclerotia, controlling this pathogen presents significant challenges [26]. Microsclerotia serve as dormant structures enabling the fungus to persist in the soil without a host for an extended period. Upon exposure to root exudates, these microsclerotia undergo germination, subsequently infiltrating the roots to initiate new infection cycles. Therefore, microsclerotia represent the predominant source of *V. dahliae* infection. Over time, the molecular mechanisms underlying Verticillium wilt have gradually been elucidated through the identification of genes implicated in virulence, growth, conidiation, and development [27]. *V. dahliae* is known to form an interaction interface with its host, developing a penetration peg to penetrate the epidermis of plant roots [28]. Given the demonstrated connections between ubiquitin ligase E3s and plant pathogens, investigating their role in *V. dahliae* can help uncover the pathogenic mechanisms of this fungus. Our study screened for pathogenicity-related ubiquitin ligase E3 in *V. dahliae* and discovered that the knockout of *VdBre1* (*VDAG_09768*) [29] and *VdHrd1* (*VDAG_04599*) compromised the pathogenicity of *V. dahliae*. The focus of this paper is to illuminate the potential roles of these two genes in the pathogenic mechanism of *V. dahliae*.

## 2. Materials and Methods

### 2.1. Fungal Strains and Cultivation Procedures

The v991 strain, a defoliating strain of *V. dahliae* (kindly provided by Professor Guiliang Jian, Institute of Plant Protection, Chinese Academy of Agricultural Sciences), was employed as the wild type for the study. The fungal strains used in this investigation, inclusive of both solid and liquid potato glucose mediums, were cultivated at a maintained temperature of 25 °C. For the purpose of this study, five mutant strains were labeled as *ΔVDAG_04599* (*ΔVdHrd1*), *ΔVDAG_09768* (*ΔVdBre1*), *ΔVDAG_00389*, *ΔVDAG_07200*, and *ΔVDAG_07898*. These were complemented by *ΔVdBre1com* and *ΔVdHrd1com*, with the wild type denoted as WT. The competent strains of *Escherichia coli* DH5a and *Agrobacterium tumefaciens* AGL1 (Shanghai Weidi Biotechnology, Shanghai, China) were employed for plasmid construction and fungal transformation, respectively.

### 2.2. Sequence Alignment, Phylogenetic Analysis, and Structural Modeling

Muscle, operating with default parameters, was employed to carry out multiple sequence alignments. The construction of the neighbor-joining (NJ) phylogenetic tree was facilitated via MEGA X v10.2, employing the following parameters: a WAG + F + G substitution model, pairwise deletion, and bootstrap (with 1000 replicates). 

### 2.3. Constructing Vectors and Transforming Fungi

A gene-replacement plasmid was generated by amplifying two homologous arms from the genomic DNA of *V. dahliae* strain v991, using specified primers. Additionally, the hygromycin resistance (Hygr) cassette was amplified from the psilent1 vector. These three fragments were ligated to the linearized pGFP plasmid, followed by selection of transformants using designated 500 μg/mL cefotaxime and 50 μg/mL hygromycin B. Gene-deleted strains were subsequently screened in accordance with our previously established protocol.

To assemble the gene complementation vector, the complete gene, including its native promoter (1500 bp) and terminator sequences (1000 bp), was cloned into the pG418 complementation plasmid, and introduced into the relevant gene deletion strains (*ΔVdBre1* and *ΔVdHrd1*). The selection of transformants and the generation of complementation strains were performed using 500 µg/mL cefotaxime, 50 µg/mL hygromycin B, and 25 µg/mL G418, and confirmed via RT-PCR (Appendix A).

Assessment of root colonization ability was performed by amplifying eGFP and integrating it into plasmid pG418, which was then introduced into the *ΔVdBre1*, *ΔVdHrd1*, and WT strains. To observe the penetration structure, VdSep5-GFP expression strains were produced by amplifying VdSep5 from v991 cDNA and ligating it into the linearized pG418-pGPDA-GFP-TrpC vector. Subcellular localization of VdBre1 and VdHrd1 was investigated by amplifying the full coding sequence of each from v991 cDNA and fusing it with eGFP fragments, thus creating the respective VdBre1-eGFP or VdHrd1-eGFP vectors. Further investigation involved introduction of the pG418-mCherry-NLS vector into the VdBre1-eGFP strain. Transformation of all vectors was conducted in *V. dahliae* using *Agrobacterium tumefaciens* AGL1, in line with our previously published methodology [27].

### 2.4. Pathogenicity Analysis

The influence of specific genes on pathogenicity was investigated using spores derived from WT, *ΔVdBre1*, *ΔVdHrd1*, *ΔVdBre1com*, *ΔVdHrd1com*, and sterile water as a control, to inoculate four-week-old *Gossypium hirsutum* plants. The root dip-inoculation method was utilized, each pot was poured with 20 mL *V. dahliae* spore suspension (1 × 10^7^ spores/mL), then transferred into a plant cultivation chamber set at 26 °C for a 16 h:8 h light–dark cycle. All pathogenicity assays were independently performed thrice. 

### 2.5. Root Colonization Analysis

To observe pathogenic processes, cotton plants were cultivated in 1/2 MS medium for two weeks. Following this, roots of the plants were immersed in a suspension of *V. dahliae* spores (10^5^ spores/mL) from WT/eGFP, *ΔVdBre1*/eGFP, and *ΔVdHrd1*/eGFP strains for 10 min, followed by a gentle rinse and replanting on the 1/2MS medium. Post-inoculation after 48 h, roots were examined under a confocal laser microscope with a 40× objective lens, utilizing a 488 nm laser for GFP excitation. Fluorescent images were subsequently captured (LSM900, Carl Zeiss, Jena, Germany).

### 2.6. Observation of Penetration Pegs and Septin Rings

Following sterilization, the cellophane membrane was positioned onto a minimal medium (NaNO_3_ 6 g/L, KH_2_PO_4_ 1.52 g/L, KCl 0.52 g/L, MgSO_4_·7H_2_O 0.52 g/L, L-glutamic acid 2.9426 g/L, pH 7.2, agar 15 g/L) and inoculated with a 2 × 2 mm fungal sample, followed by incubation at 25 °C for three days. The membrane with the developed colonies was cautiously arranged on a glass slide for examination of the penetration pegs and septin rings using an optical microscope (DM3000, Carl Zeiss, Jena, Germany).

### 2.7. Staining of Lipid Droplets and the Endoplasmic Reticulum (ER)

Strains WT, *ΔVdBre1*, and *ΔVdBre1com* were cultivated on a solid PDA medium at 25 °C over a five-day period. The harvested mycelium and spores were subsequently processed for staining. To visualize lipid droplets, spores were rinsed in sterile water and stained using a diluted Nile red stain solution (SL0201, Coolaber, Beijing, China) with a working concentration of 0.1 mg/mL for 10 min at 25 °C in the dark. Following the staining process, spores were washed in a PBS buffer and observed under a microscope. The emitted red fluorescence signal was inspected through a fluorescence microscope with 63× objective lenses (DM3000, Carl Zeiss, Jena, Germany). 

In order to stain the ER, the *V. dahliae* strains were cultured on solid potato dextrose agar (PDA) medium at 25 °C for a period of 5 days. The mycelium was subsequently collected and subjected to staining procedures for visualization. The ER was stained with ER-Tracker Blue-White DPX (Thermo Fisher, Waltham, MA, USA) according to the manufacturer’s protocol. A stock solution of ER-Tracker Blue-White DPX (1 mM) was diluted in Hanks’ balanced salt solution to achieve a final concentration of 500 μM. The mycelium was stained with the working staining buffer for 30 min at 37 °C in the dark. Prior to observation, the mycelium was rinsed with Hanks’ balanced salt solution. Fluorescent images of ER-Tracker Blue-White DPX and eGFP were obtained using a confocal microscope (Olympus, Tokyo, Japan). During microscopy, 374 nm and 488 nm lasers were used to excite the ER-Tracker Blue-White DPX and eGFP, respectively. 

### 2.8. RNA Sequencing

In order to conduct transcriptome profiling analyses, all strains (WT, *ΔVdBre1*, and *ΔVdHrd1*) were cultivated for 10 days on a cellophane membrane placed on top of a potato dextrose agar medium. Total RNA was subsequently isolated with the use of TRIzol reagent (Invitrogen Life Technologies, Carlsbad, CA, USA). RNA samples were then evaluated for concentration, quality, and integrity using a NanoDrop spectrophotometer (Thermo Scientific, Waltham, MA, USA) prior to transcriptome sequencing. Following this, 3 µg of RNA was utilized in the construction of sequencing libraries via a multifaceted process encompassing the purification of mRNA through poly-T oligo-attached magnetic beads, fragmentation by divalent cations within an Illumina proprietary fragmentation buffer, first-strand cDNA synthesis using random oligonucleotides and SuperScript II, creation of second-strand cDNA via DNA polymerase I and RNase H, and conversion of the leftover overhangs into blunt ends through exonuclease/polymerase activities, concurrent with the removal of associated enzymes. This process was finalized by the adenylation of the 3′ termini of DNA fragments, followed by ligation with Illumina PE adapter oligonucleotides, facilitating subsequent hybridization. The AMPure XP system (Beckman Coulter, Beverly, CA, USA) was employed for purification to selectively target cDNA fragments of a length ranging between 400 and 500 bp. DNA fragments that were ligated with adaptor molecules at both ends underwent selective amplification using an Illumina PCR Primer Cocktail within a 15 cycle PCR reaction. The purified end products (AMPure XP system) were quantified via an Agilent high-sensitivity DNA assay and processed on a Bioanalyzer 2100 system (Agilent, Santa Clara, CA, USA). Ultimately, the sequencing libraries were processed on the NovaSeq 6000 platform (Illumina, San Diego, CA, USA) by Shanghai Personal Biotechnology Cp. Ltd. Clean reads extracted from the raw transcriptome sequence data were aligned to the reference genome of *V. dahliae*, forming the basis for subsequent analyses.

### 2.9. Analysis of Differentially Expressed Genes (DEGs), Gene Ontology (GO), and Kyoto Encyclopedia of Genes and Genomes (KEGG)

The software Tbtools v2.003 was implemented to discern DEGs between the wild-type and the mutant strains. In this context, genes showcasing a fold change equal to or exceeding 2, coupled with a *p*-value of less than 0.01, were classified as significant DEGs. The DEGs thus identified were functionally categorized in accordance with the results derived from GO annotations (http://geneontology.org/ (accessed on 14 July 2020)) and KEGG annotations (https://www.genome.jp/kegg/ (accessed on 15 July 2020)).

### 2.10. Statistical Analysis

The collected data were articulated as the mean value ± standard error of the mean. A one-way analysis of variance (ANOVA) independent-samples Tukey’s range test was conducted for data analysis, facilitated by the software GraphPad Prism 8.0 (San Diego, CA, USA). Statistical significance was attributed to a *p*-value of less than 0.05. To indicate levels of significance, asterisks were employed, with a single asterisk (*) signifying a *p*-value of less than 0.05, two asterisks (**) denoting a *p*-value of less than 0.01, three asterisks (***) denoting a *p*-value of less than 0.001, and four asterisks (***) denoting a *p*-value of less than 0.0001. 

### 2.11. RT-PCR and RT-qPCR Assay

Total RNA was isolated using the OminiPlant RNA Kit (CWBIO, Taizhou, China), followed by reverse transcription with the PrimeScript RT Reagent Kit (TaKaRa Dalian Biotechnology, Dalian, China). The subsequent RT-PCR was carried out to analyze the expression of *VdBre1* and *VdHrd1*, featuring an initial denaturation at 94 °C for 2 min, and 25 cycles of 94 °C for 30 s, 57 °C for 30 s, and 72 °C for 30 s. The 18sRNA gene of *V. dahliae* served as the control. Appendix A provide details of the primers and transformants used, respectively.

The study proceeded to assess the expression levels of genes from the transcriptome results through the use of RT-qPCR. The total RNA from strains (WT, *ΔVdBre1*, *ΔVdHrd1, ΔVdBre1com*, and *ΔVdHrd1com*) was individually isolated utilizing the OminiPlant RNA Kit (CWBIO, China), followed by reverse transcription with the PrimeScript RT Reagent Kit (TaKaRa Dalian Biotechnology, Dalian, China). For the purposes of this investigation, the 18sRNA gene served as the reference control. A comprehensive list of all primers used throughout the study can be found in the Appendix A.

### 2.12. Western Blot Assays 

In order to detect H2B monoubiquitination and H3K4 trimethylation, Western blot assays were conducted on WT, *ΔVdBre1*, and *ΔVdBre1com* strains. These were cultured in PDA medium for seven days at 25 °C prior to protein extraction. The detection of H2B ubiquitination was performed using the ubiquityl-histone H2B antibody (Beyotime, Shanghai, China). Tri-methyl-histone H3 (Lys4) rabbit polyclonal antibody, and histone H3 rabbit polyclonal antibody (Beyotime, Shanghai, China) were utilized at a dilution ratio of 1:1000, adhering to standard Western blot methodologies.

## 3. Result

### 3.1. Identification of Ubiquitin Ligase E3 in V. dahliae

A total of fifteen putative ubiquitin ligase E3s were identified in the annotated genome of *V. dahliae* v991 and examined via a Conserved Domain Database (CDD) search in NCBI and smart alignment. The results revealed that all the scrutinized candidate proteins exhibited the characteristic domain of ubiquitin ligase E3 (Figure 1). Gene structure analysis output disclosed the presence of varied quantities of introns and exons within all 15 genes (Appendix A), encoding a range of 273 to 3973 amino acids and predicting molecular weights that ranged from 30.5 kDa to 440.4 kDa. The phylogenetic scrutiny of *V. dahliae*’s ubiquitin ligase E3 via MEGA7.0 segregated E3s primarily into two branches: the HECT family (positioned at the C-terminal of the protein) and the RING family, validating the conserved domain results (Figure 1). It is of note that VDAG_07200 has been categorized within the RING family based on phylogenetic analysis; however, the presence of a RING domain has not been identified so far.

### 3.2. Exploration of RING E3 Ligases Associated with Pathogenicity

To explore the correlation between RING E3 ligase genes and the pathogenicity of *V. dahliae*, knockout and pathogenicity analyses were performed on cotton plants. Via gene knockout, five mutants emerged from RING E3 ligase genes, specifically *ΔVDAG_07200*, *ΔVDAG_07898*, *ΔVDAG_00389*, *ΔVDAG_04599*, and *ΔVDAG_09768* (Appendix A). Both the wild-type (WT) and the knockout mutant strains were inoculated on 4-week-old cotton using conidial suspensions (10^7^ spores/mL), with sterile water serving as the control. Data analysis unveiled that WT, *ΔVDAG_07200*, *ΔVDAG_07898*, and *ΔVDAG_00389* elicited typical necrotic lesions on cotton. Conversely, plants infected with *ΔVDAG_04599* and *ΔVDAG_09768* mutants did not manifest Verticillium wilt symptoms, analogous to the sterile water control (Figure 2A,B and Appendix A). Following this, complementary assays utilizing *ΔVd04599com* and *ΔVd09768com* strains re-established full virulence (Figure 2B), signified by vascular system discoloration (Figure 2C). As a whole, the performed experiments identified the *VDAG_04599* and *VDAG_09768* genes as indispensable to the pathogenicity of *V. dahliae*.

### 3.3. Sequence Features and Subcellular Localization of VdBre1 and VdHrd1

Sequence analysis indicated that *VDAG_09768* encodes 687 codons and *VDAG_04599* encodes 785 amino acids, with both genes possessing four exons and three introns. VDAG_09768 incorporates a Bre1 domain and a RING domain (Appendix A); hence, the associated protein is named VdBre1. Meanwhile, VDAG_04599 contains an Hrd1 domain (Appendix A), resulting in the name VdHrd1. Multiple sequence alignment confirmed the extensive conservation of Bre1 and Hrd1 domains across diverse fungi (Appendix A). 

Predictive subcellular localization pointed to the existence of a nuclear localization signal sequence, EDRKRPAISGADDIAPPSKRHQ, located between the 19th and 40th amino acid of VdBre1 (Figure 3A), and five transmembrane domains in VdHrd1 (Figure 3B). To validate these predictions, VdBre1 and VdHrd1 were tagged with eGFP and co-transformed with mCherry-NLS or underwent ER-Tracker Blue-White DPX staining, respectively. Co-localization of green and red fluorescence was detected in the co-transformed strains of WT/VdBre1-eGFP and WT/VdBre1-eGFP/mCherry-NLS, suggesting that VdBre1 is localized at the nucleus (Figure 3A). The overlap of ER-Tracker Blue-White DPX staining and eGFP fluorescence in WT/VdHrd1-eGFP strains was consistent with VdHrd1 being localized to the ER in *V. dahliae* (Figure 3B). Compared with the WT strain and complemented strain, the *VdBre1* and *VdHrd1* mutants showed slower hyphal growth, and the introduction of *VdBre1-eGFP* and *VdHrd1-eGFP* genes partially restored the phenotype of the mutants (Appendix A).

### 3.4. VdBre1 and VdHrd1 Deletion Impaired Penetration in V. dahliae

In order to examine the differential root invasion tendencies between the WT and *ΔVdBre1*/*ΔVdHrd1* mutants in *V. dahliae*, we developed eGFP-labeled mutants of these two mutants. Considering the pivotal role that roots play as the entry point for *V. dahliae* into the vascular bundles of host plants, 2-week-old cotton roots were inoculated with *ΔVdBre1*/eGFP, *ΔVdHrd1*/eGFP and WT/eGFP strains, respectively. At 48h post-inoculation, while the WT/eGFP strain successfully colonized xylem vessels, *ΔVdBre1*/eGFP and *ΔVdHrd1*/eGFP remained limited to the peripheral region of the roots, with no presence detected in the xylem vessels (Figure 4A). 

Noting that the penetration peg is a crucial component of *V. dahliae*’s host invasion strategy, we investigated the effects of *VdBre1* and *VdHrd1* deletion on the formation of the penetration peg. Strains of WT, *ΔVdBre1*, *ΔVdHrd1*, *ΔVdBre1com*, and *ΔVdHrd1com* were inoculated onto a cellophane membrane overlaid on MM medium and subsequently analyzed through microscopic inspection after a three-day incubation period. Our observations revealed that while the WT, *ΔVdBre1com*, and *ΔVdHrd1com* produced a well-structured penetration peg, the *ΔVdBre1* and *ΔVdHrd1* strains failed to develop a similar structure (Figure 4B). The VdSep5 protein is of considerable importance for *V. dahliae*’s penetration peg formation. We transferred VdSep5-eGFP into the WT, *ΔVdBre1*, and *ΔVdHrd1* strains and assessed septin ring formation. While the WT strain exhibited a compact and regularly arranged ring structure, *ΔVdBre1* and *ΔVdHrd1* failed to produce any septin formation (Figure 4C). Thus, we concluded that the absence of *VdBre1* or *VdHrd1* impaired the formation of the penetration structure, thereby obstructing the fungal colonization of the plant vascular bundles.

### 3.5. RNA Sequencing Analyses Unveil the Roles of VdBre1 and VdHrd1 in V. dahliae

To delineate the influence of *VdBre1* deletion on gene expression, we performed transcriptome-level analysis contrasting WT and *ΔVdBre1* strains. This revealed 442 genes as down-regulated, and 395 genes as up-regulated in the *ΔVdBre1* strain. The function of these differentially expressed genes was explored through the Kyoto Encyclopedia of Genes and Genomes (KEGG) pathway analysis, uncovering significant enrichment in pathways related to carbon metabolism, lipid metabolism, amino acid metabolism, and energy metabolism (Figure 5A). Selected genes from the lipid metabolism pathway were validated using qPCR, displaying significant down-regulation of *VDAG_07507*, *VDAG_09868*, *VDAG_02241*, and *VDAG_07771*, whereas *VDAG_05865*, *VDAG_07080*, *VDAG_00583*, *VDAG_07881*, and *VDAG_07695* were markedly up-regulated (Figure 5B,C). 

The genes examined in this study are detailed further in the Appendix A. Given the notable difference observed in the expression of genes associated with the lipid metabolism pathway, we conducted a visualization of the lipid droplets of *V. dahliae* using Nile red staining. This revealed the presence of large round lipid droplets predominantly localized in the WT strain, while the lipid droplets in the *ΔVdBre1* strain appeared to be smaller and dispersed more loosely (Figure 5D,E).

In an effort to elucidate the transcriptional implications of *ΔVdHrd1*, we engaged in an analysis of differentially expressed genes adhering to the stringent criteria of an absolute log2 value greater than or equal to 2 and a *p* value less than 0.01. This analysis yielded a total of 578 up-regulated and 417 down-regulated genes. Remarkably, the genes that were up-regulated were largely characterized by their association with molecular chaperones and folding catalysts and endoplasmic reticulum physiological processes (Appendix A). On the other hand, down-regulated genes were predominantly enriched in pathways linked to ascorbate and aldarate metabolism (Appendix A).

### 3.6. VdBre1 Governs H2B Ubiquitination and H3K4 Trimethylation

The existing literature has cited the impact of Bre1 on histone H2B ubiquitination in yeast. Consequently, we evaluated the ubiquitination levels in WT, *ΔVdBre1*, and *ΔVdBre1com* strains. The data pointed to a significant decline in ubiquitin levels in *ΔVdBre1* compared to WT and *ΔVdBre1com* strains (Figure 6A). Histone H2B ubiquitination is reportedly critical for H3K4 methylation [30]. An assessment of H3K4 trimethylation levels illuminated a significant drop in trimethylation levels of the *ΔVdBre1* strains (Figure 6A). Given the crucial role both H2B ubiquitination and H3K4 trimethylation play in the expression of subsequent secreted proteins [23], we inspected the expression of cell-wall-degrading enzymes in *V. dahliae* using qPCR. This examination disclosed a rise in the expression of cell-wall-degrading enzyme genes (*VDAG_02709*, *VDAG_05344*, and *VDAG_06165*) (Figure 6B). In summary, our findings suggest that VdBre1 plays an indispensable role in both H2B ubiquitination and H3K4 trimethylation pathways in *V. dahliae*.

## 4. Discussion

Recent investigations have underscored the crucial part ubiquitination plays in enhancing plant disease resistance and in the pathogens’ ability to suppress a plant’s innate immunity. Specifically, the fungal effector can destabilize plant PTI by targeting the degradation of the host’s ubiquitin ligase E3, while the host’s ubiquitin ligase E3 can degrade the fungal effector, thus reducing its impact on the host’s PTI [31]. Notably, the effector AVR3a in *Phytophthora infestans* has been shown to interact with and modify CMPG1, a potato ubiquitin ligase E3, to prevent its degradation, resulting in aberrant INF1-triggered cell death [32]. However, the ubiquitin ligase E3′s role in virulence regulation varies and is dependent on different pathways in diverse pathogenic fungi. For instance, ubiquitin ligase E3 is involved in virulence regulation via appressorium development in *M. oryzae* [9], whereas in *F. graminearum*, it regulates secondary metabolism [33]. Our present study emphasizes the identification of two ubiquitin ligases E3, namely, VdBre1 and VdHrd1, in relation to the pathogenicity of *V. dahliae*. Interestingly, many studies have identified processes related to ubiquitin ligase E3 as potential therapeutic targets in several diseases. For example, chlorofusin was discovered by blocking the MDM2-mediated ubiquitination of P53 via inhibition of protein interactions due to overexpression of MDM2/HDM in certain cancers [34]. Nevertheless, research into inhibiting ubiquitin ligase E3 in pathogenic fungi remains underexplored.

The ability of many phytopathogens to infiltrate host plants via penetration structures, such as appressoria in *M. oryzae* [9], is well-documented. In the vicinity of the appressorial pore in infected cells, an extensive circular F-actin network is found, with a particular reorientation of the actin cytoskeleton facilitating successful plant infection [35,36]. Septins, which are instrumental in relocating and reorganizing the cytoskeleton to determine cell shape, are co-localized with the F-actin network at the appressorium pore in *M. oryzae* [36]. The formation of a septin ring is crucial for supporting the creation of an F-actin ring network at the appressorium base prior to plant infection [37]; the septin ring also acts as a diffusion barrier, governing the lateral diffusion of membrane-associated proteins implicated in plant infection. Our investigation’s findings showed that deletion of *VdBre1* and *VdHrd1* impaired the development of both the penetration peg and septin ring, thus blocking colonization of the host plant’s vascular bundles. Statistical analysis revealed that after three days of cultivation, 96 ± 19 (n = 7), penetration pegs could be counted in the sparse mycelium region around the wild-type strain colony periphery, while no penetration pegs were observed in the *ΔVdBre1* and *ΔVdHrd1* mutant strain in this region. However, penetration pegs in the dense mycelium region in the center of the colony could not be clearly distinguished, making it impossible to quantify the number of penetration pegs in the wild-type strain or to determine if the mutant strain possesses penetration pegs. Nevertheless, it can be inferred from these results on penetration pegs at the colony periphery that the deletion of *VdBre1* and *VdHrd1* has impaired the ability of *V. dahliae* to form penetration pegs. 

In addition, transcriptomic analysis indicated the involvement of VdBre1 in lipid metabolism, and observations showed that deletion of *VdBre1* affected the formation of concentrated lipid droplets, which are crucial for generating adequate swelling pressure to penetrate the plant tissue surface and are vital for the pathogen’s virulence [38]. Multiple previous studies have highlighted the importance of Bre1 in relation to the ubiquitination of histone H2B and H3K4 trimethylation [23]. Consistent with prior findings, our results demonstrated that *VdBre1* deletion led to a substantial reduction in both H2B ubiquitination and H3K4 trimethylation. Secondary metabolic biosynthesis genes are typically arranged in secondary metabolite gene clusters (SMGCs), and chromatin structure and post-translational histone modifications are significant factors in SMGC regulation. H3K4 is a renowned histone post-translational modification associated with the promoter region of actively transcribed genes, and the regulation of the secondary metabolite system controls the production of plant cell-wall-degrading enzymes (PCWDEs) in response to changing nutritional conditions [39]. Previous research has proposed that Bre1 positively regulates the expression of fungal effectors. However, this study revealed that three cell-wall-degrading enzymes (PCWDEs) were up-regulated in *ΔVdBre1*, suggesting a feedback mechanism in response to the pathogenicity caused by the loss of *VdBre1*, aligning with the absence of trimethylation of H3K4 in *A. fumigatus*, but increased activation and expression of secondary metabolites [40]. However, this did not compensate for the loss of virulence, underscoring the diverse functions of Bre1 across different species.

The transcriptomic results highlighted the primary involvement of *VdHrd1* in regulating ER-related processes. Molecular chaperones, including Hsp40, Hsp70, and Hsp90, play a critical role in assisting the appropriate folding of proteins during ER stress [38,39,41]. Furthermore, we found that deletion of the *VdHrd1* gene led to a significant up-regulation in the expression of genes involved in molecular chaperoning and folding catalysis. Our findings also demonstrated that the expression of a homologous gene of KAR2 [42], VDAG_05516, in *V. dahliae*, which responds positively to ER stress in yeast, was upregulated 7-fold. Interestingly, contrary to *M. oryzae* [43] and *A. fumigattus* [44], we did not observe a synergistic effect of VdHrd1 and VdDer1 in response to ER stress induced by tunicamycin (TM), suggesting the involvement of alternative pathways in *V. dahliae*’s response to this stress (unpublished). HRD1 has been identified as a regulator of cholesterol production through the modulation of the turnover of the rate-limiting enzyme HMGCR (HMG-CoA reductase) in yeast. Subsequent investigations have demonstrated the pivotal involvement of HRD1 in the ER-associated degradation of misfolded/unfolded proteins, thereby safeguarding cells from ER-stress-induced apoptotic cell death [45]. Nevertheless, our empirical findings are insufficient to provide substantial evidence for the localization of VdHrd1 within the ER.

In conclusion, we have successfully identified two crucial ubiquitin ligases, VdBre1 and VdHrd1, that play instrumental roles in the virulence of *V. dahliae* and are localized in the nucleus and ER, respectively. In addition, both enzymes exert a regulatory influence on the development of the penetration peg and the formation of the septin ring. Specifically, VdBre1 is involved in H2B ubiquitination, H3K4 trimethylation, and lipid metabolism, whereas VdHrd1 plays a role in regulating genes related to molecular chaperone and folding catalyst pathways.

## Figures and Tables

**Figure 1 jof-09-01037-f001:**
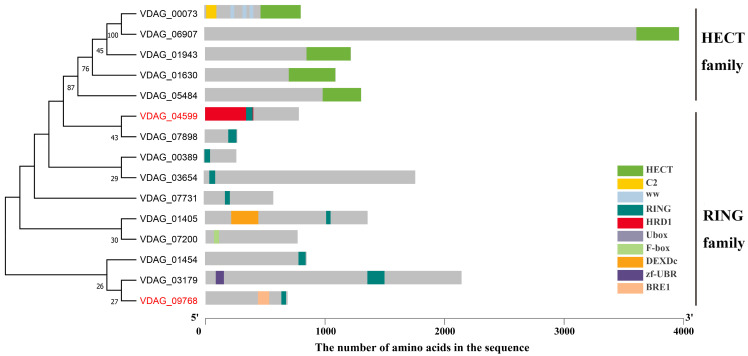
Phylogenetic analysis and conservation domains of ubiquitin ligase E3 proteins with full-length amino acid sequence identified in *Verticillium dahliae*. The tree was constructed using the neighbor joining method by MEGA X. The bootstrap values were annotated in the evolutionary tree. On the right are the schematics of the conserved domain. HECT, homologous to the E 6-AP carboxyl terminus domain; RING, really interesting new gene domain; C2, protein kinase C conserved region 2 domain; WW, two conserved tryptophans domain; HRD1, HMG-CoA reductase degradation protein 1 domain; F-box, F-box protein domain; DEXD, conserved domain with DExD sequence; Zf-UBR, zinc finger in N-recognin domain; BRE1, histone H2B ubiquitination domain. VDAG_07200 has not exhibited the presence of a RING domain. The two genes highlighted in red were found to be associated with the pathogenicity of *V. dahliae* in this study.

**Figure 2 jof-09-01037-f002:**
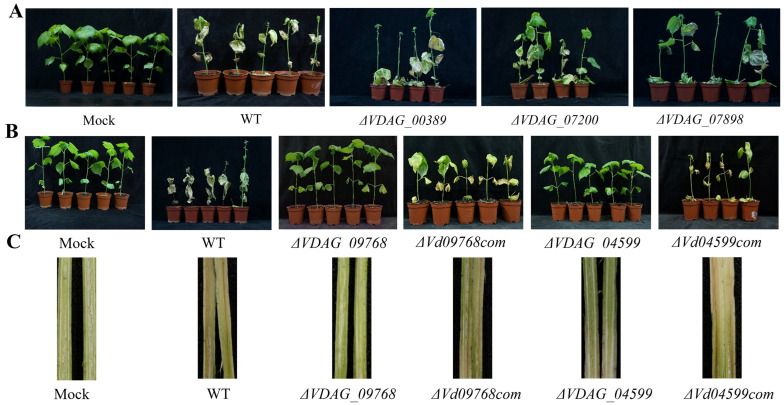
Assessment of pathogenicity on cotton plants. (**A**) The WT, along with *ΔVDAG_07200*, *ΔVDAG_07898*, and *ΔVDAG_00389* strains, elicited typical necrotic lesions on cotton plants. (**B**) Conversely, the *ΔVDAG_04599* and *ΔVDAG_09768* mutant strains failed to manifest symptoms of Verticillium wilt, bearing similarity to the sterile water control. However, upon reintroduction of *VDAG_04599* and *VDAG_09768*, the mutants reverted to full virulence. *ΔVDAG_04599com* and *ΔVDAG_09768com* are complemented strains. (**C**) Vascular browning was evident in the stems of plants inoculated with WT, *ΔVDAG_04599com*, and *ΔVDAG_09768com*. Sterile water treatments were used as control (mock). Photographs were taken 20 days post-inoculation, and the experiment was repeated three times.

**Figure 3 jof-09-01037-f003:**
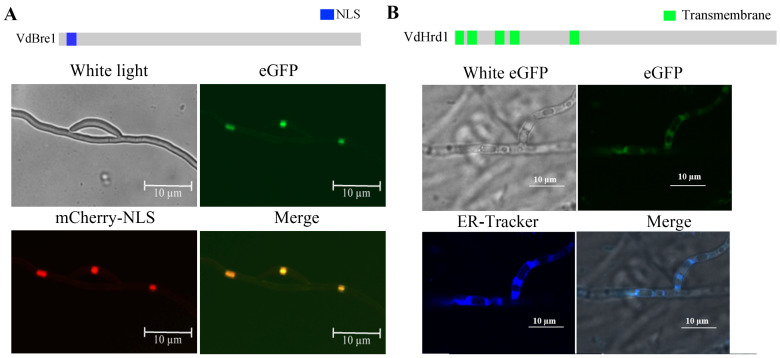
Assessment of subcellular localization of *VdBre1* and *VdHrd1* in *Verticillium dahliae*. (**A**) The green fluorescence corresponds to the expression of *VdBre1* tagged with GFP. Red fluorescence indicates mCherry-NLS, a nuclear localization signal. The merged images exhibit a significant co-localization of VdBre1 and mCherry-NLS within the nucleus. (**B**) The green signal indicates the expression of VdHrd1 tagged with GFP, while the blue signal represents the ER as indicated by ER-Tracker Blue-White DPX staining. Overlaid images (merge) demonstrate co-localization of VdHrd1-GFP with the ER. Scale bars correspond to 10 µm.

**Figure 4 jof-09-01037-f004:**
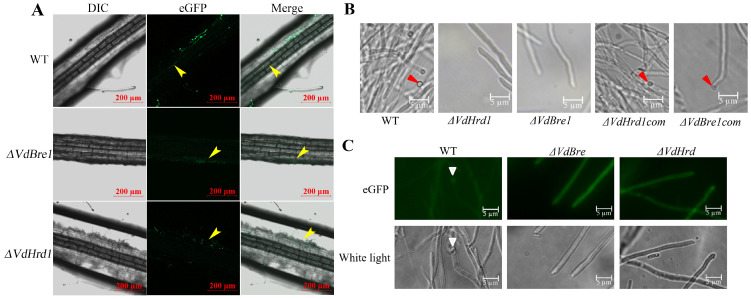
Root colonization and penetration analyses of *Verticillium dahliae*. (**A**) Invasion and colonization following the inoculation of 2-week-old cotton roots with WT/eGFP, *ΔVdBre1*/eGFP, and *ΔVdHrd1*/eGFP strains. Yellow arrows indicate hyphae of *V. dahliae*. (**B**) Penetration pegs formed on the cellophane membrane by wild-type (WT), *ΔVdBre1*, *ΔVdBre1com*, *ΔVdHrd1* and *ΔVdHrd1com* strains. Red arrows indicate penetration pegs. (**C**) Cellular localization of VdSep5-GFP in WT, *ΔVdBre1 and ΔVdHrd1* during development of the penetration peg. The white arrow indicates the septin ring.

**Figure 5 jof-09-01037-f005:**
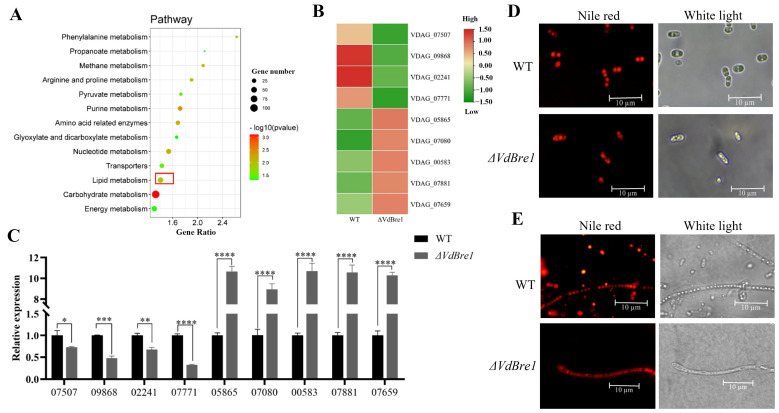
Regulatory role of VdBre1 in lipid-metabolism-related gene expression (**A**). KEGG enrichment analysis for differentially expressed genes. Genes enriched in lipid metabolism pathways are denoted by red rectangle. Translation: The red box represents the enriched lipid metabolism-related gene. A heatmap for DEGs associated with lipid metabolism from transcriptomic data. Red represents relatively higher expression, while green represents relatively lower expression (**B**); these results were subsequently verified by RT-qPCR analyses (**C**). The lipid droplet staining of spores (**D**) and hyphae (**E**) in WT and *ΔVdBre1*, respectively. Values represent the mean ± standard deviation of three independent replications. Significant differences, denoted by *, **, ***, and ****, represent *p*-values less than 0.05, 0.01, 0.001, and 0.0001, respectively.

**Figure 6 jof-09-01037-f006:**
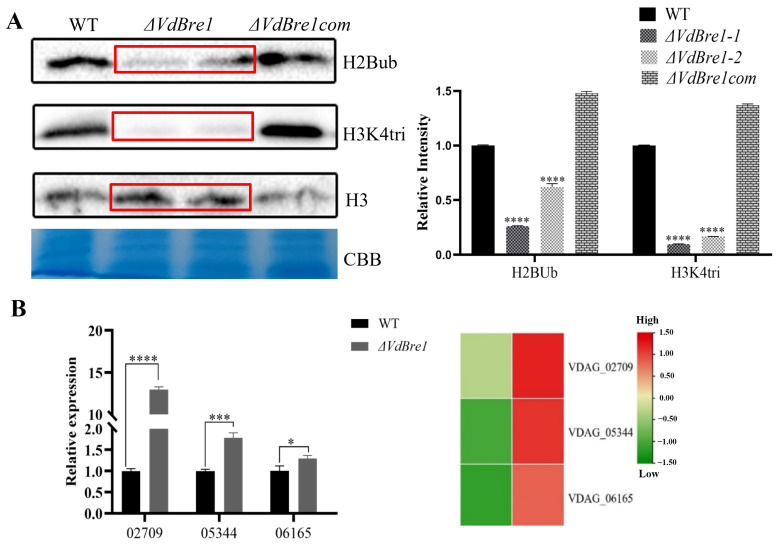
The involvement of VdBre1 in the regulation of H2B ubiquitination (H2Bub) and H3K4 trimethylation (H3K4tri). (**A**) This figure presents a Western blot analysis aimed at examining H2B ubiquitination and H3K4 trimethylation; their relative intensity was evaluated using Image J v. 1.42 (on the (**right**)). The red box represents the results of the mutant samples. (**B**) This heatmap illustrates the transcriptional profiling of cell-wall-degrading enzyme genes from the transcriptomic dataset. Red represents relatively higher expression, while green represents relatively lower expression (on the (**right**)). The corresponding RT-qPCR analyses, presented on the left, subsequently validated these results. Values (A and B) represent the mean ± standard deviation of three independent replications. Significant differences, denoted by *, ***, and ****, represent *p*-values less than 0.05, 0.001, and 0.0001, respectively.

## Data Availability

The RNA-Seq sequence dataset supporting the results of this study is available at NCBI with accession numbers SAMN35653360, SAMN35653361, SAMN35653362, SAMN35653363, SAMN35653364, SAMN35653365.

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
