# Peer review of "Functional Role of RING Ubiquitin E3 Ligase VdBre1 and VdHrd1 in the Pathogenicity and Penetration Structure Formation of Verticillium dahliae"

_jof, 2023, doi:10.3390/jof9101037_

Round 1

Reviewer 1 Report

The research article “ Functional Role of RING Ubiquitin E3 Ligase VdBre1 and VdHrd1 in the Pathogenicity and Penetration Structure For mation of Verticillium dahliae “ highlighted  the function of ubiquitin ligase E3 in V. dahliae and offered fresh theoretical perspectives. The authors claimed their research identifies two novel virulence-related genes and partially explicates their roles in virulence-associated structures and gene regulatory pathways. These findings augment our understanding of the molecular mechanisms inherent to V. dahliae.

The Abstract of the research paper is appropriate and comprehensive

The Introduction portion needs improvement; there must be more description of the pathogen, virulence, and disease development.

Material methods: Provides satisfactory description of the methodologies adopted. However, strain acquisition, multiplication, and then inoculation need more details.

The host plant where the strains were isolated, the spore concentration was inoculated, rest is ok.

Results are described well and supported with good discussion. The conclusion is satisfactory.    

Author Response

Reviewer 1

Q1—The Introduction portion needs improvement; there must be more description of the pathogen, virulence, and disease development.

Answer: We briefly introduced Verticillium wilt in p2L90-96, and also added more information related with the infection process.

Q2—Material methods: Provides satisfactory description of the methodologies adopted. However, strain acquisition, multiplication, and then inoculation need more details.

Answer: More information regarding the strains has been incorporated.

Q3—The host plant where the strains were isolated, the spore concentration was inoculated, rest is ok.

Answer: This fungus has been extensively utilized in various laboratories in China for several decades, and it is indeed difficult to trace its original isolation. We provide the laboratory from which we obtained this fungus in M&M. The spore concentration was added in the pathogenicity testing experiment.

Reviewer 2 Report

In this manuscript, the authors conducted reverse genetic analysis of RING type E3 ubiquitin ligases in V. dahlia. They generated knockout mutants of five RING type E3 genes, and VdBre1 as well as VdHrd1 were identified to be required for the full virulence of V. dahliae. Defects of the penetration peg and septin ring were observed in the knockout mutants, which supported the compromised pathogenicity. Similar to the yeast Bre1, VdBre1 was also involved in the H2B ubiquitination and methylation of H3K4 in V. dahliae, which resulted in different expression of some pathogenicity related genes. Additionally, evidence on subcellular localization of both E3 ligases were provided. Furthermore, RNA -seq performed on VdBre1 and VdHrd1 mutants suggested that these two E3 ligases are involved in the fungal metabolic pathways. Overall, the study showed adequate evidence of the roles of VdBre1 and VdHrd1 on pathogenicity of V. dahliae, which is consistent with their analogues in other fungal organism.

However, I have several concerns about this study.

1.      The yeast analogue of VdBre1, has been known to be required for cell size control and development. In the manuscript, evidence suggested that VdBre1 is involved in pathogenicity. However, the contribution of VdBre1 in fungal pathogenicity may be indirect. Most DEGs in ΔVdBre1 are categorized into metabolism pathways, the mutant may also defect in fungal development. Authors should characterize more on both VdBre1 and VdHrd1 knockout mutants, to test whether the mutants are defective in growth and fungal development.

2.      Section 3.2, five E3 genes were selected to generate loss-of-function mutants. However, like in Figure 2, only one single knock-out line of each gene is shown. To confirm the mutant phenotype, two independent lines of each gene are required. Or else a complementing line should be included as a control.

3.      How FM4-64 staining is performed is missing in Methods.

4.      Alignment of the RING domains should be performed to see whether the E3 core residues are conserved. Are they real E3s? In vitro ubiquitination assay should be performed to confirm that are indeed E3s.

In addition, there are some issues about writing and figure annotations, including but not limited to the list as below. I suggest authors do a thorough proofreading on the manuscript to improve them.

1.      Figure 1, it will be helpful to mention whether phylogenetic tree is constructed with full length protein sequence or DNA sequence.

2.      Figure 1, annotations of the scale bars are missing; bootstrap values of each branch should be provided to support phylogenetic tree.

3.      Line 229, “Figure 1B”, however, there is no Figure 1B in Figure 1. Figure labelling needs to be improved.

4.      Figure 5A and 5B, annotations of scale bars of are missing.

5.      Line 324, “..” should be “.”.

6.      Figure 5E, the photo of ΔVdBre1, taken in a field full of mycelia, is messy and blurry, which make it difficult to conclude. A photo taken with less mycelia (like the one of WT) will be better.

7.      Line 368, “Fig. 6C” should be “Fig. 6B”. Same as that in Line 375.

See above

Author Response

Reviewer 2

Q1—The yeast analogue of VdBre1, has been known to be required for cell size control and development. In the manuscript, evidence suggested that VdBre1 is involved in pathogenicity. However, the contribution of VdBre1 in fungal pathogenicity may be indirect. Most DEGs in ΔVdBre1 are categorized into metabolism pathways, the mutant may also defect in fungal development. Authors should characterize more on both VdBre1 and VdHrd1 knockout mutants, to test whether the mutants are defective in growth and fungal development.

Answer: Our observations revealed that the mutant form of this gene, compared to the wild type, exhibited only a slightly slower growth rate, with no apparent differences in morphological characteristics such as mycelial spores. Considering the goal of maintaining a concise article, we chose not to include the growth experimental results in the initial draft. In the new version, we have created a supplementary file containing the results of the colony growth, which have also been included in the article. Figure S5

Q2—Section 3.2, five E3 genes were selected to generate loss-of-function mutants. However, like in Figure 2, only one single knock-out line of each gene is shown. To confirm the mutant phenotype, two independent lines of each gene are required. Or else a complementing line should be included as a control.

Answer: As shown in Figure S2, each gene underwent at least two mutations, followed by two initial pathogenicity tests. After identifying the mutants with virulence defects, gene complementation was performed on the mutants of virulence defects, followed by another pathogenicity test to confirm the restoration of virulence. Finally, we confirmed the association of the two genes with virulence. Due to space limitations, we did not present additional figures. In the new version, we have included more pathogenicity test results in the supplementary file. Figure S3

Q3—How FM4-64 staining is performed is missing in Methods.

Answer: The method has been inserted. P4L175-183

Q4—  Alignment of the RING domains should be performed to see whether the E3 core residues are conserved. Are they real E3s? In vitro ubiquitination assay should be performed to confirm that are indeed E3s.

Answer: RING and HECT are representative functional domains of two classes of E3s, and E3s do not have completely overlapping functional domains. We have previously attempted ubiquitination experiments in Escherichia coli, but without success.

Q5—Figure 1, it will be helpful to mention whether phylogenetic tree is constructed with full length protein sequence or DNA sequence.

Answer: The relevant information has been included in the legend of Figure 1.

Q6— Figure 1, annotations of the scale bars are missing; bootstrap values of each branch should be provided to support phylogenetic tree.

Answer: The annotations of the scale bars and bootstrap values have been added.

Q7—Line 229, “Figure 1B”, however, there is no Figure 1B in Figure 1. Figure labelling needs to be improved.

Answer: The error has been corrected.

Q8—Figure 5A and 5B, annotations of scale bars of are missing.

Answer: The annotations of scale bars have been added.

Q9—Line 324, “..” should be “.”.

Answer: The error has been corrected.

Q10—Figure 5E, the photo of ΔVdBre1, taken in a field full of mycelia, is messy and blurry, which make it difficult to conclude. A photo taken with less mycelia (like the one of WT) will be better.

Answer: We replaced the original images with clearer ones.

Q11—Line 368, “Fig. 6C” should be “Fig. 6B”. Same as that in Line 375.

Answer: The error has been corrected.

Reviewer 3 Report

Xing Yang and colleagues present an article on the identification and initial characterization of two RING-domain containing E3 ubiquitin ligase enzymes in the fungus Verticillium dahliae. The cellular localization is monitored by fluorescence microscopy and the effects of enzyme knock-out are associated with pathogenicity. An involvement of one of these ubiquitin ligases with the regulation of expression of histone H2B is demonstrated. Finally, an expression analysis at a transcriptome level is performed, revealing interconnections with central metabolic pathways.

In general, the article may contain some interesting and important information, however, in my opinion, in its current form the information presented appears rather preliminary and predictable and less original, moreover, certain conclusions are not fully supported by the results.

Lines 24-27: This article is not elucidating the function of E3 enzymes, but identifies and characterizes two RING-type E3 homologues. As also cited elsewhere in the text (Ref. 14), relation to pathogenicity is also not entirely novel.

Lines 33-40 could be placed after the general information on E3 enzymes.

Lines 56-70: More information should be given in relation to other characterized enzymes in fungi. For example, RING type ligases have also been studied in detail in Aspergilli. In general, this part of the introduction deserves some elaboration.

Lines 211-216: Please include relevant information on the anti-H2Bub1 antibody used in this study.

Figure 2: A growth test analysis of the knock out mutants compared to the wt is necessary. If growth of the knock out mutants is severely affected or delayed, then this would probably be reflected in the phenotypes related to pathogenicity, making any pathogenicity impairment rather predictable (see also below in the comment on Figures 4B,C).

Line 281 and elsewhere: Figure 3A shows that there is a functional NLS in Bre1. This only suggests (and not corroborates) that VdBre1 is localized at the nucleus.

Figure 3B: FM4-64 in other fungi is used as a marker for intracellular structures related to endosomes and endocytosis in general. Even in terminal staining times (over 30min) only late endosomes and vacuolar membranes are stained. Here the authors state that the red signal from FM4-64 staining represents the endoplasmic reticulum. Does this dye stain different structures in this fungus? Moreover, what seems to fluoresce in Figure 3B is the plasma membrane and some foci resembling vacuoles. Do the authors mean that the signal colocalizes predominantly at the plasma membrane (as also indicated in the discussion, Line 443)? In general, the quality of these images should be improved, the colocalizations should also be performed with other dyes or known organelle markers and be quantified. In addition, a growth test should be shown to prove that the GFP-tagged Hrd1 is fully functional.

Figure 4B and 4C: It appears as if the mycelium in the knock out mutants is less dense compared to the wt. Could the absence of penetration pegs and septin rings in the knock-out mutants be simply explained by a general growth defect or delay in these mutants? How many hyphae were observed in this case? Longer incubation times should also be monitored in this occasion. All these results should also be quantified.

Author Response

Reviewer 3

Q1—Lines 24-27: This article is not elucidating the function of E3 enzymes, but identifies and characterizes two RING-type E3 homologues. As also cited elsewhere in the text (Ref. 14), relation to pathogenicity is also not entirely novel.

Answer: We attempted to investigate the protein interaction of VdBre1 and VdHrd1 in depth using techniques such as IP-MS and proximity labeling. However, the protein cannot be detected in western blot analysis, which limits further experimentation.

Q2—Lines 33-40 could be placed after the general information on E3 enzymes.

Answer: The sentence order has been adjusted.

Q3—Lines 56-70: More information should be given in relation to other characterized enzymes in fungi. For example, RING type ligases have also been studied in detail in Aspergilli. In general, this part of the introduction deserves some elaboration.

Answer: We have incorporated additional research findings on E3 genes in fungi into the Introduction section. P2L59-64, L68-77

Q4—Lines 211-216: Please include relevant information on the anti-H2Bub1 antibody used in this study.

Answer: The information has been added.

Q5—Figure 2: A growth test analysis of the knock out mutants compared to the wt is necessary. If growth of the knock out mutants is severely affected or delayed, then this would probably be reflected in the phenotypes related to pathogenicity, making any pathogenicity impairment rather predictable (see also below in the comment on Figures 4B,C).

Answer: Our observations revealed that the mutant form of this gene, compared to the wild type, exhibited only a slightly slower growth rate, with no apparent differences in morphological characteristics such as mycelial spores. Considering the goal of maintaining a concise article, we chose not to include the growth experimental results in the initial draft. In the new version, we have created a supplementary file containing the results of the colony growth, which have also been included in the article. Fig S5

Q6—Line 281 and elsewhere: Figure 3A shows that there is a functional NLS in Bre1. This only suggests (and not corroborates) that VdBre1 is localized at the nucleus.

Answer: The inaccurate expression has been modified.

Q7—Figure 3B: FM4-64 in other fungi is used as a marker for intracellular structures related to endosomes and endocytosis in general. Even in terminal staining times (over 30min) only late endosomes and vacuolar membranes are stained. Here the authors state that the red signal from FM4-64 staining represents the endoplasmic reticulum. Does this dye stain different structures in this fungus? Moreover, what seems to fluoresce in Figure 3B is the plasma membrane and some foci resembling vacuoles. Do the authors mean that the signal colocalizes predominantly at the plasma membrane (as also indicated in the discussion, Line 443)? In general, the quality of these images should be improved, the colocalizations should also be performed with other dyes or known organelle markers and be quantified.

Answer: FM4-64 staining only confirms localization on the intracellular membrane. Therefore, we have adjusted our statement accordingly. Due to the fact that a substantial amount of literature suggests the localization of HRD1 in the endoplasmic reticulum (ER), we focused on this localization. However, the co-localization of HRD1 with other proteins did not yield satisfactory results, prompting us to employ staining methods instead.

Q8—In addition, a growth test should be shown to prove that the GFP-tagged Hrd1 is fully functional.

Answer: We have included the experimental results regarding the growth of GFP-tagged Hrd1 strains in the supplementary materials. Fig S5

Q9—Figure 4B and 4C: It appears as if the mycelium in the knock out mutants is less dense compared to the wt. Could the absence of penetration pegs and septin rings in the knock-out mutants be simply explained by a general growth defect or delay in these mutants? How many hyphae were observed in this case? Longer incubation times should also be monitored in this occasion. All these results should also be quantified.

Answer: After gene knockout, the growth of mutants on the basal culture medium is slower, but not completely inhibited. In the revised version, the results of the growth experiment are provided as supplementary information. In this cultivation period, there were already sufficient hyphae available for observation of penetration pegs. In fact, the wild type could already be observed penetration pegs at an earlier time. The observation of penetration pegs does not rely on larger colonies.

It is challenging to observe penetration pegs in densely populated areas of colonies; therefore, observations are limited to the colony periphery. For the wild-type, there is a significant variation in the number of observed penetration pegs between agar plates, and the overall quantity is also limited (2-10 / each field of view), making it challenging to quantify accurately.

Round 2

Reviewer 2 Report

The authors have addressed most of my concerns.

Author Response

Thank your review.

Reviewer 3 Report

The revised manuscript has been improved. However, in my opinion, authors’ responses of in certain queries require more elaboration, further text additions and additional experiments.

Concerning reviewer’s Q7: Authors have now changed the statement and talk about intracellular membranes. However, as mentioned before, FM4-64 staining is a dynamic process that labels the entire intracellular endocytic pathway (plasma membrane, early and late endosomes, and vacuolar membranes) according on the staining period. Why did the authors choose to stain for 15min, where most probably only the initial endocytic stages, including the plasma membrane would be stained? In the provided image in Figure 3B, although the image quality is less optimal (the hypha appears out of focus and the culture contaminated?), not only intracellular membranes are stained, but also the plasma membrane and certain spots that resemble vacuoles. Some of those signals, including the plasma membrane signal, appear to overlap at both the GFP and RFP channels. Therefore, talking about intracellular membranes is in this case too generalized and should be avoided. Authors should either provide new higher quality image, or even better a series of FM4-64 staining pictures in different staining times (eg from 10min to 60min) and compare the possible overlap with Hrd1. This is particularly important since the tagged strains do not fully complement the Hrd1 deletion, raising the possibility that they are partially misfolded, leading to their accumulation in vacuolar compartments. On the other hand, if much less overlap with FM4-64 is observed during the entire endocytic process, then this may well indicate that Hrd1 is localized at intracellular membranes related the ER, or the peripheral ER, a result in agreement with observations in other organisms.

Concerning reviewer’s Q9: I respect the authors’ reply that the quantification of these phenotypes is challenging. However, if quantification is challenging, then the conclusions based on the interpretation of results are also challenging. Thus, emphasis should in this case be lowered and this limitation in the quantification should be stated and discussed in a relevant place of the text.

Author Response

Q1: Concerning reviewer’s Q7: Authors have now changed the statement and talk about intracellular membranes. However, as mentioned before, FM4-64 staining is a dynamic process that labels the entire intracellular endocytic pathway (plasma membrane, early and late endosomes, and vacuolar membranes) according on the staining period. Why did the authors choose to stain for 15min, where most probably only the initial endocytic stages, including the plasma membrane would be stained? In the provided image in Figure 3B, although the image quality is less optimal (the hypha appears out of focus and the culture contaminated?), not only intracellular membranes are stained, but also the plasma membrane and certain spots that resemble vacuoles. Some of those signals, including the plasma membrane signal, appear to overlap at both the GFP and RFP channels. Therefore, talking about intracellular membranes is in this case too generalized and should be avoided. Authors should either provide new higher quality image, or even better a series of FM4-64 staining pictures in different staining times (eg from 10min to 60min) and compare the possible overlap with Hrd1. This is particularly important since the tagged strains do not fully complement the Hrd1 deletion, raising the possibility that they are partially misfolded, leading to their accumulation in vacuolar compartments. On the other hand, if much less overlap with FM4-64 is observed during the entire endocytic process, then this may well indicate that Hrd1 is localized at intracellular membranes related the ER, or the peripheral ER, a result in agreement with observations in other organisms.

Answer: Thank you for your valuable suggestion. Following your suggestion, we performed FM4-64 staining at different time intervals (5 min - 60 min, with a 5 min interval). Initially, we could only observe FM4-64 signal on the cell membrane. After 10 minutes, FM4-64 signal was primarily localized in membrane structures of the cytoplasm. By 25 minutes, we mainly observed red fluorescent FM4-64 signal on the vacuolar membrane and cell membrane. It is important to note that after 20min staining, the green fluorescent signal of Hrd1-eGFP was weak. We are unsure if prolonged FM4-64 staining could affect the stability of the Hrd1-eGFP protein. This is why we initially opted for a 15-minute staining time. After all, even with the same staining time, there are variations in the degree of staining between hyphae. Hence, the co-localization results of FM4-64 merely indicate the localization of Hrd1-eGFP within membranous structures of the cytoplasm. However, no green fluorescence signal was observed in the vacuole in different staining time.

Based on our difficulties in accurately determining the finer details of the fluorescent signal using a conventional fluorescence microscope, we used a laser scanning confocal microscope in another university. Considering the potential misunderstandings associated with FM4-64. Based on literature reports (DOI: 10.1371/journal.ppat.1005793), we purchased ER-Tracker Blue-White DPX, which has been reported to provide better endoplasmic reticulum (ER) staining results in V. dahliae. We subsequently stained the ER using this reagent, and it showed good co-localization with our protein.

Q1: Concerning reviewer’s Q9: I respect the authors’ reply that the quantification of these phenotypes is challenging. However, if quantification is challenging, then the conclusions based on the interpretation of results are also challenging. Thus, emphasis should in this case be lowered and this limitation in the quantification should be stated and discussed in a relevant place of the text.

Answer: We re-cultivated the strains and quantified the number of penetration pegs. The strains were cultured on cellophane membrane, and small pieces of cellophane membrane with strains were cut and reversed to observe the penetration pegs on the side in contact with the culture medium. However, this count is only for the sparsely populated area around the colony, as there are too many hyphae in the middle of the colony. Additionally, the poor translucency of light also affects observation. We also discussed related issues in the discussion sectin. “Statistical analysis revealed that after three days of cultivation, 96±19 (n=7) penetration pegs could be counted in the sparse mycelium region around the wild-type strain colony periphery, while no penetration pegs were observed in the ΔVdBre1 and ΔVdHrd1 mutant strain in this region. However, penetration pegs in the dense mycelium region in the center of the colony could not be clearly distinguished, making it impossible to quantify the number of penetration pegs in the wild-type strain or determine if the mutant strain possesses penetration pegs. Nevertheless, it can be inferred from these results on penetration pegs at the colony periphery that the deletion of VdBre1 and VdHrd1 has impaired ability of V. dahliae to form penetration pegs.”